# Systematic identification of needlefish (Belonidae) species using molecular genetic and morphological markers in the Mediterranean and Black Seas

Cemal Turan[ID]¹*, Petya Pavlova Ivanova², Servet Ahmet Doğdu[ID]¹,³, Deniz Ergüden[ID]¹, Violin Stoyanov Raykov², Maria Yankova², Deniz Yağlioğlu⁴, Ayşegül Ergenler¹

1 Molecular Ecology and Fisheries Genetics Laboratory, Faculty of Marine Sciences and Technology, Iskenderun Technical University, Iskenderun, Hatay, Türkiye, 2 Marine Biology and Ecology Department, Institute of Oceanology, Bulgarian Academy of Sciences, Varna, Bulgaria, 3 Iskenderun Technical University, Maritime Technology Vocational School of Higher Education, Underwater Technologies, Iskenderun, Hatay, Türkiye, 4 Department of Biology, Faculty of Arts and Sciences, Duzce University, Duzce, Türkiye

* cemal.turan@iste.edu.tr

## Abstract

In this study, we aimed to clarify the taxonomic status of Belonidae species distributed in the Mediterranean Sea and the Black Sea by conducting detailed genetic and morphological markers. A total of 550 needlefish samples were caught between January 2022 and January 2024. The data set used in the study contains a total of 171 sequences for the *COI* gene and 120 sequences for the *12s rRNA* gene from different Belonidae species, including data from GenBank. Systematic analysis of needlefish species was investigated by using sequencing of mtDNA *COI* and *12s rRNA* gene regions and morphological characters in the Turkish Marine Waters. A separate analysis of the two mitochondrial genes supported by morphological characters revealed that each species is grouped within itself. The genetic and morphological analyses showed that *Belone belone acus* and *Belone belone euxini* which are considered as the subspecies of *Belone belone* are not subspecies of the genus *Belone* and should be considered at the species level, *Belone belone*. *Belone svetovidovi* is also considerably different from *Belone belone* and should be considered as a different species. *T. acus imperialis*, which is thought to be distributed in the Mediterranean Sea, is not a subspecies of *Tylosorus acus and should be revised as Tylosorus imperialis* which genetically differs from *Tylosorus acus* and also other *Tylosorus* species at the species level.

## Introduction

The family Belonidae, occurring in all tropical and temperate seas, is characterized by their long bodies, long beaks with sharp teeth, and the position of the dorsal and anal fins relative to each other and to the caudal fins, and represents 10 genera and 46 valid species worldwide [1].

**Data availability statement:** All genetic data set files are available from the NCBI (GenBank) database (COI gene region Belone belone : PQ549693-PQ549712, Tylosorus imperialis : PQ549719-PQ549748, Belone svetovidovi: PQ549749-PQ549768, PQ550612-PQ550631, PQ550664-PQ550683, PQ554004-PQ554013, 12s rRNA gene region: Belone belone, PQ564486-PQ564495, Belone svetovidovi : PQ564532-PQ564601, Tylosorus imperialis: PQ564605-PQ564634). Also, the file with accession numbers of all samples is stored in the zenodo database with the doi: 10.5281/zenodo.14414507. The morphological dataset used in the study is stored in the zenodo database with doi: 10.5281/zenodo.14249768. Morphological data can be accessed here.

**Funding:** This study was supported by the Scientific and Technological Research Council of Turkey (TÜBİTAK-121N777) and the Bulgarian Academy of Sciences (BAS-IC-TR/1/2022-2023). The authors thank TUBITAK for their support. The funders had no role in study design, data collection and analysis, decision to publish, or preparation of the manuscript.

**Competing interests:** The authors have declared that no competing interests exist.

The family are epipelagic, feeding near the surface and has a migratory pattern similar to the mackerel species [2–3]. Four species, *Belone belone, B. svetovidovi*, *Tylosurus imperialis* and *T. choram*, are distributed in the Mediterranean, while only two of them are distributed in the Black Sea [3–6]. The garfish *Belone belone* (Linnaeus, 1760) is distributed in the eastern Atlantic Ocean and the Mediterranean Sea [7]. Until now, three subspecies have been identified by Collette and Parin [8]. Of these species, *B. belone belone* is distribution the northeastern Atlantic. *B. belone gracilis* is distributed from the south of France in the Mediterranean Sea to the Canary Islands in the Atlantic. *B. belone euxini* is distributed in the Black Sea and the Sea of Azov [3,9]. The short-beaked garfish *Belone svetovidovi*, which is benthic fish species, is most commonly observed in tropical waters. The distribution range of this species in the eastern Atlantic Ocean spreads as far as Southern Ireland, Spain, Portugal, Israel and Türkiye. [3,10,11]. In recent years, it has also been reported to be distributed in the Marmara Sea and the Black Sea [6].

The needlefish *Tylosurus acus* is a species that inhabits epipelagic waters globally, distributed across tropical and subtropical regions [12], having a considerable number of subspecies [7,12]. Collette and Parin [13] identified five subspecies of *Tylosurus acus*; *Tylosurus acus acus* (Lacepède 1803) in the western Atlantic, *T. a. imperialis* (Rafinesque 1810) in the Mediterranean Sea. Furthermore, *T. a. rafale* Collette & Parin 1970 is reported in the Gulf of Guinea, *T. a. melanotus* (Bleeker 1850) along the Indo-western Pacific and extending into the eastern Pacific, and *T. a. pacificus* (Steindachner 1876) in the eastern Pacific which was subsequently revised to species level as *Tylosurus pacificus* [14].

There are limited number of studies on the biological parameters of these species classified as species or subspecies level in the Black and Aegean Seas; *Belone belone* [9,15–19], *Belone belone euxini* [20]. Rare studies have been carried out on the biology of *Belone svetovidovi* [21] and *Tylosurus acus imperialis* [17,22]. There have also been limited genetic studies on the phylogeny of the Belonidae family. Lovejoy [23] studied the phylogenetic relationships among different genera of needlefishes, sauries, halfbeaks and flying fishes using mitochondrial, nuclear and morphological characters to better understand the evolutionary changes in the jaw ontogeny of needlefishes. Collete and Banford [14] compared *Tyleserus* genera with *ATPase 8,6* and *cyt b* gene regions in their study of Pacific stingfish populations and found that the subspecies *Tylosurus acus pacificus* differed morphologically and genetically from *Tylosurus acus*, and concluded that *T. pacificus* should be registered as a species. Imsiridou et al. [12] carried out the genetic and phenotypic identification of *Tylosurus acus imperialis* specimens from the North Aegean Sea using *16s rRNA* gene region and reported that *T. a. imperialis* is genetically and morphologically different from *T. choram* and *T. crocodilus*. Yankova et al. [9] analysed the genetic and morphometric characteristics of the *Belone belone* population on the Nesebar coast of Bulgaria and reported that *B. belone* experienced a population decline due to overfishing on the Bulgarian coast which was genetically different from the Turkish population [19].

Phylogenetic analyses of fish species are fundamental to taxonomy as they facilitate the understanding and classification of evolutionary relationships among fish species. Phylogenetic analyses can include data from three different domains: morphological (phenotypic characteristics), molecular (genetic structures) and behavioural [24]. Through these analyses, fish species can be grouped with similar characteristics, and phylogenetic trees can be created to determine the evolutionary relationships among different species [25]. Therefore, the phylogenetic analyses enable us to make accurate classifications in fish taxonomy and better understand the relationships among species.

In this study, it was aimed to clarify the taxonomic status of Belonidae species distributed in the Mediterranean and Black Sea by conducting genetic and morphological markers.

## Materials and methods

### Sampling

A total of 550 needlefish samples were caught from commercial trawl and purse seine fisheries between January 2022 and January 2024. Of these putative sampling, 50 samples were *B. belone* (25 Nesebar and 25 Akçakoca), 350 samples were *B. svetovidovi* (50 Muğla, 50 İzmir, 50 Çanakkale, 50 Yalova, 50 Düzce, 50 Sinop and 50 Rize coasts) and 150 samples were *T. imperialis* species (İskenderun, Mersin and Antalya coasts) (Fig 1). All samples were examined morphologically, and muscle and/or fin tissues were taken for genetic analyses. Identification of the specimens was done following Collete & Parin [8]. Sampling studies were carried out within the scope of the permission given by the General Directorate of Fisheries and Aquaculture of the Ministry of Agriculture and Forestry of the Republic of Türkiye. This study does not require Local Ethics Committee Approval as experimental animals were not used in this study.

### Genetic Analysis

The specimens were transported to the laboratory and stored at a temperature of -30 °C in a deep freeze until the DNA extraction process was initiated. Total genomic DNA was extracted from muscle or fin tissues using a modified phenol-chloroform isoamyl alcohol method [26]. The mtDNA *COI* and *12s rRNA* gene regions were amplified through PCR with universal primers (Table 1) [27–28].

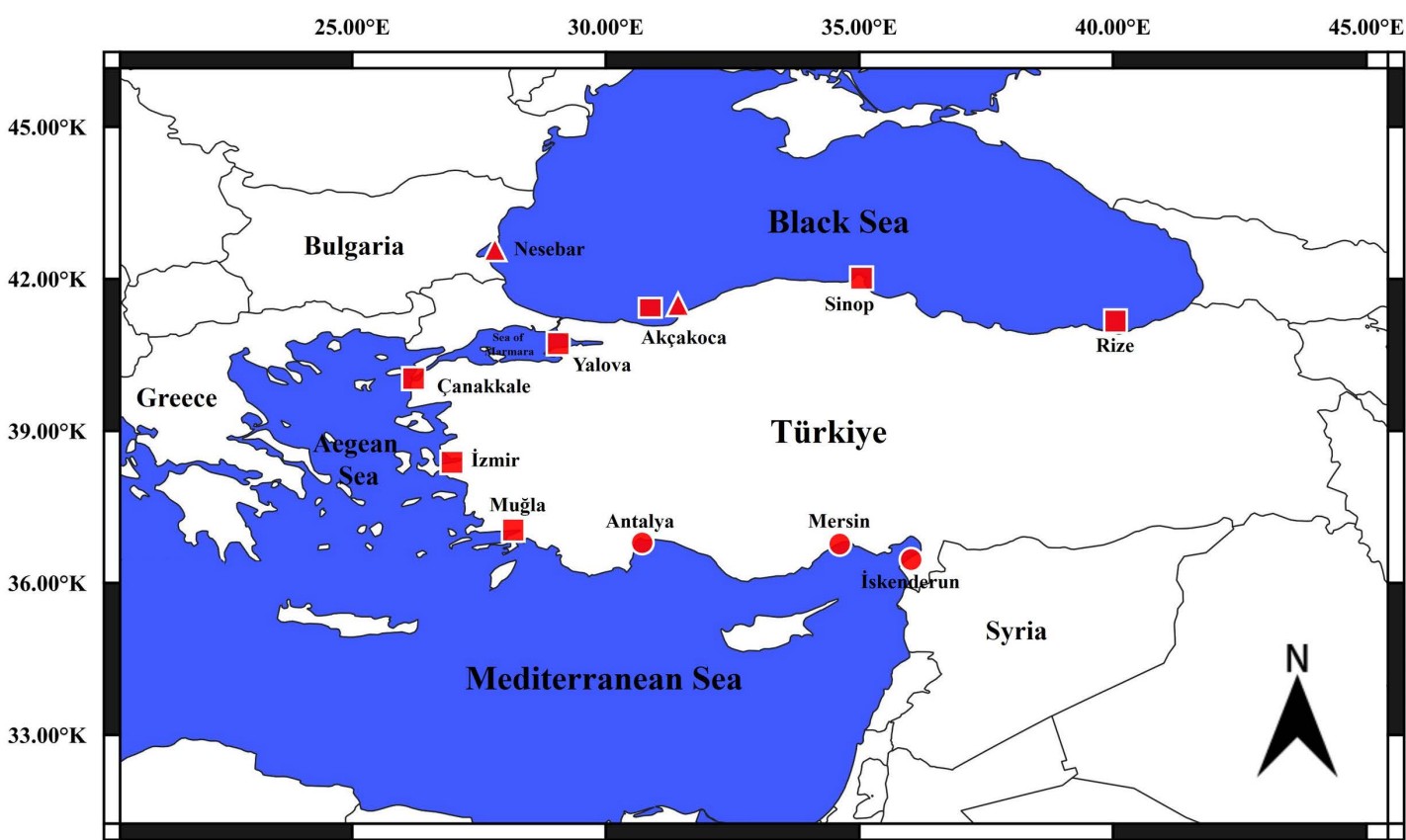

**Fig 1. Sampling location. Circle:** *T. imperialis*, **Squere:** *Belone svetovidovi*, **Trigle:** *Belone belone*.

**Table 1.  Universal primers of *COI* and *12s rRNA* gene regions.**

| COI | COI_F: | 5'-TCA ACC AAC CAC AAA GAC ATT GGC AC-'3' |
|---|---|---|
| | COI_R: | 5'-ACT TCA GGG TGA CCG AAG AAT CAG AA-'3' |
| **12s rRNA** | 12s_F: | 5'- CAA ACT GGG ATT AGA TAC CCC ACT AT -3' |
| | 12s_R: | 5'- GAG GGT GAC GGG CGG TGT GT -3' |

The PCRs were carried out in a 50 μl total volume with 0.4 ɥM of each primer, 0.2 mM of dNTP and 1 U of Taq DNA polymerase in a PCR buffer that comprise 20 mM of Tris–HCl (pH 8.0), 1.5 mM of MgCl$_2$, 15 mM of KCl and 1.5 μl template DNA. The denaturation step was at 94°C for 30 s, 60°C for 30 s, and 75°C for 45 s for 35 cycles followed by a final extension for 7 min at 75 °C., then the PCR products were screened using electrophoresis on 1.5% agarose gel. DNA sequencing was carried out by chain termination method by Sanger et al. [29] with Bigdye Cycle Sequencing Kit V3.1 and ABI 3130 XL genetic analyzer. The DNA sequence analyses were performed on 10 sample specimens of various sizes from each population. The partial mtDNA sequences were initially aligned with the Clustal W [30], and the final alignment was finalized manually with BioEdit [31]. In statistical analyses, haplotypes diversity, genetic diversity [32], and the genetic differences [33] were found using the program of Arlequin [34]. Haplotypes were generated by the DnaSP 6 program [35]. The minimum spanning tree (MST) of haplotypes was constracted with the Popart [36–37].

For *COI* gene regions, a total of 120 specimens, 20 *B. belone* specimens from two populations, 70 *B. svetovidovi* specimens from seven populations, and 30 *T. imperialis* specimens from three populations, were sequenced for the mtDNA *COI* gene region. For *12s rRNA* gene regions, a total of 110 specimens, 10 *B. belone* samples from one population, 70 *B. svetovidovi* specimens from seven populations, and 30 *T. imperialis* specimens from three populations, were sequenced for the mtDNA *12s rRNA* gene region. The data set used in the study contains a total of 171 sequences for the *COI* gene and 120 sequences for the *12s rRNA* gene from different Belonidae species, including data from NCBI (S1 Table). Furthermore, the species of *Ablennes hians* (Valenciennes, 1846) was used as an outgroup as it belongs to the Belonidae family. The sequences obtained in our study were uploaded to NCBI and accession numbers were obtained. Accession numbers for the *COI* gene region: *B. belone*, Nesebar and Akcakoca PQ549693-PQ549712, *B. svetovidovi* Rize and Sinop PQ549749-PQ549768, Akcakoca and Yalova PQ550612-PQ550631, Çanakkale and Izmir PQ550664-PQ550683, Muğla PQ554004-PQ554013, *Tylosorus imperialis* İskenderun, Antalya and Mersin PQ549719-PQ549748. Accession numbers for the 12s rRNA gene region: *B. belone*, PQ564486-PQ564495, *B. svetovidovi* PQ564532-PQ564601, *Tylosorus imperialis* PQ564605-PQ564634 (S2 Table). Phylogenetic relationships were estimated with the MEGAX software [38]., using the maximum likelihood (ML) and maximum parsimony (MP) phylogenetic tree analyses.

## Morphological analysis

Morphometric measurements were carried out on the photographs of the fish samples using the BioMorph software [39]. The morphometric characters analysed are presented in Fig 2.

A principal component analysis (PCA) and discriminant function analysis (DFA) were performed for the classification of species based on morphometric characters. The majority of observed variability in morphological characters can be attributed to differences in size [41]. Accordingly, shape analysis must be conducted in a manner that is not influenced by size variation to prevent misinterpretation of the results [42]. Therefore, the first principal component

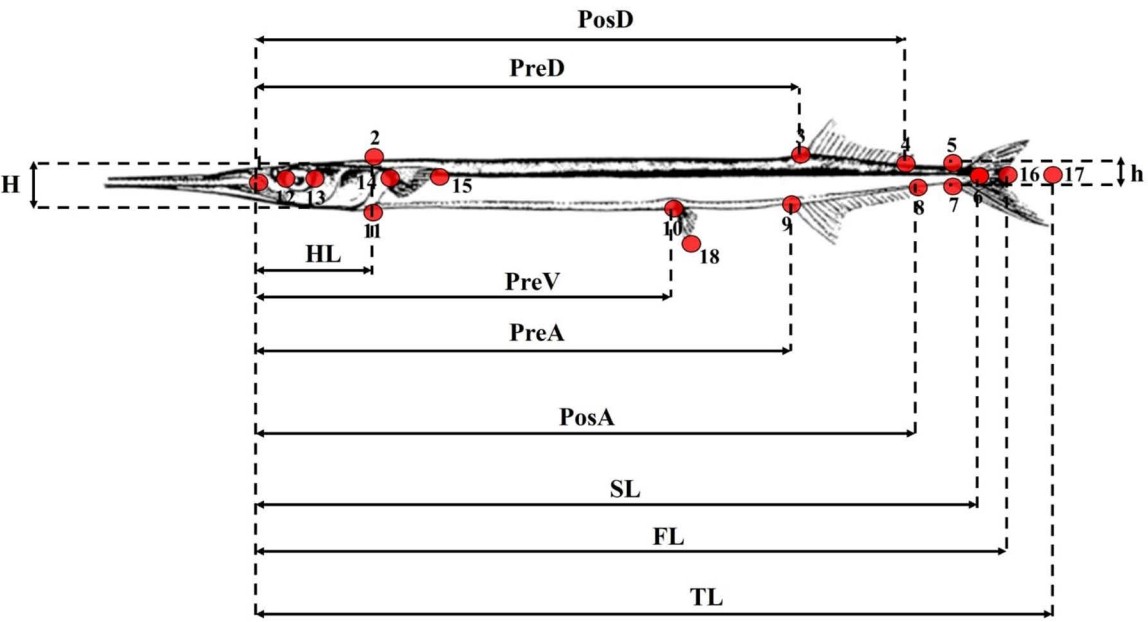

**Fig 2. Morphometric measurements of needlefish.** TL: total length, FL: fork length, SL: standard length, PreD: Predorsal distance, PostD: Postdorsal distance, PreA: preanal distance, PostA: Postanal distance, PreV: preventral distance, HL: head length, 12-13: eye diameter, H: maximum body height, h: minimum body height, 3-4: dorsal fin base length, 8-9: anal fin base length, 14-15: pectoral fin length, 10-18: ventral fin length (Figure: Collette, [40]).

of PCA was removed from the analyses since it represents a multivariate generalization of simple allometry (Jolicoeur, 1963; Klingenberg, 1996), subsequent components were used for multivariate morphometric analysis (S3 Table). A Hierarchical cluster analysis using Euclidean distance was performed to see morphological relationships among species. All calculations were performed in SPSS, SYSTAT and R Studio programs.

## Results

### COI gene

Polymorphisms within the mtDNA *COI* gene region were obtained as 113 variable and 493 conservative nucleotides of which 106 were parsimony informative over 606 bp sequences. The best model (HKY + G) was provided by the MEGAX software (Hall, 2013). The parameters of this model were: nucleotide frequencies A: 26.2%, T: 33.0%, C: 25.10%, G: 16.10%, + G: 0.19. The estimated Transition/Transformation bias (R) was found to be 2.07.

Twenty-eight haplotypes were found out of 120 sequences, and there were no common haplotypes between the species (Table 2). Haplotype diversity was found to be 0.8594. The minimum spanning tree (MST) analysis revealed a high amount of mutation between the species (Fig 3).

The mean differentiation ($F_{ST}$) between individuals of a species was found to be the lowest in *B. belone* and the highest in *B. svetovidovi* (Table 3). In pairwise comparison of species, the $F_{ST}$ values ranged from 0.0345 to 0.1699 (Table 3). The lowest genetic distance was observed between *B. belone* and *B. svetovidovi* (0.0345) and the highest genetic distance was observed between *B. svetovidovi* and *T. imperialis* (0.1699) species.

The Maximum Likelihood (ML) and Maximum-Parsimony (MP) phylogenetic tree analyses revealed a similar tree topology (Fig 4 and Fig 5). The two species of the genus Belone, *B.*

**Table 2. Haplotype number and its distribution between the species for *COI* gene region** (Populations: NES: Nesebar, AKC: Akcakoca; RIZ: Rize; SIN: Sinop; YAL: Yalova, CAN: Çanakkale, IZM: Izmir, MUG: Muğla, ANT: Antalya, MER: Mersin, ISK: Iskenderun).

| Haplotypes | *B. belone* | | *B. svetovidovi* | | | | | | | *T. imperialis* | | | Total |
|---|---|---|---|---|---|---|---|---|---|---|---|---|---|
| | Black Sea | | Black Sea | | | Aegean Sea | | | | Mediterranean Sea | | | |
| | *NES* | *AKC* | *RIZ* | *SIN* | *AKC* | *YAL* | *CAN* | *IZM* | *MUG* | *ISK* | *MER* | *ANT* | |
| Hap_1 | 10 | – | – | – | – | – | – | – | – | – | – | – | 10 |
| Hap_2 | – | 10 | – | – | – | – | – | – | – | – | – | – | 10 |
| Hap_3 | – | – | 8 | 8 | 5 | 2 | 6 | 6 | 6 | – | – | – | 41 |
| Hap_4 | – | – | 2 | 2 | 1 | – | – | – | – | – | – | – | 5 |
| Hap_5 | – | – | – | – | 3 | – | – | – | – | – | – | – | 3 |
| Hap_6 | – | – | – | – | 1 | – | – | – | – | – | – | – | 1 |
| Hap_7 | – | – | – | – | – | 1 | – | – | – | – | – | – | 1 |
| Hap_8 | – | – | – | – | – | 1 | – | – | – | – | – | – | 1 |
| Hap_9 | – | – | – | – | – | 1 | – | – | – | – | – | – | 1 |
| Hap_10 | – | – | – | – | – | 1 | – | – | – | – | – | – | 1 |
| Hap_11 | – | – | – | – | – | 1 | – | – | – | – | – | – | 1 |
| Hap_12 | – | – | – | – | – | 1 | – | – | – | – | – | – | 1 |
| Hap_13 | – | – | – | – | – | 1 | – | – | – | – | – | – | 1 |
| Hap_14 | – | – | – | – | – | 1 | – | – | – | – | – | – | 1 |
| Hap_15 | – | – | – | – | – | – | 1 | – | – | – | – | – | 1 |
| Hap_16 | – | – | – | – | – | – | 2 | 3 | 2 | – | – | – | 7 |
| Hap_17 | – | – | – | – | – | – | 1 | – | 2 | – | – | – | 3 |
| Hap_18 | – | – | – | – | – | – | – | 1 | – | – | – | – | 1 |
| Hap_19 | – | – | – | – | – | – | – | – | – | 4 | – | – | 4 |
| Hap_20 | – | – | – | – | – | – | – | – | – | 1 | – | – | 1 |
| Hap_21 | – | – | – | – | – | – | – | – | – | 2 | 3 | 4 | 9 |
| Hap_22 | – | – | – | – | – | – | – | – | – | 1 | 2 | 2 | 5 |
| Hap_23 | – | – | – | – | – | – | – | – | – | 1 | 1 | – | 2 |
| Hap_24 | – | – | – | – | – | – | – | – | – | 1 | 1 | – | 2 |
| Hap_25 | – | – | – | – | – | – | – | – | – | – | 1 | – | 1 |
| Hap_26 | – | – | – | – | – | – | – | – | – | – | 1 | 1 | 2 |
| Hap_27 | – | – | – | – | – | – | – | – | – | – | 1 | 1 | 2 |
| Hap_28 | – | – | – | – | – | – | – | – | – | – | – | 2 | 2 |
| Total | 20 | | 70 | | | | | | | 30 | | | 120 |

*belone* and *B. svetovidovi*, were separated from the same branch, while the other species of the genus *Tylosurus*, *T. imperialis*, was branched as a separate branch. According to ML and MP trees, *B. belone*, *B. svetovidovi* and *T. imperialis* were clustered in different sets, and *Tylosorus imperialis* differs from other *Tylosorus* species at the species level by forming a separate branch with the *Tylosorus acus imperialis* reference sequences. These results prove that the species previously referred to as *T.a. imperialis* from the Mediterranean Sea *is Tylosorus imperialis.*

## 12s rRNA gene

After alignment, there were observed 46 variable and 379 conservative nucleotides of which 40 were parsimony informative over 425 bp sequences obtained from mtDNA *12s rRNA* gene region. The best model (K2) was provided by the MEGAX software [43]. The parameters of this model were: nucleotide frequencies A: 25%, T: 25%, C: 25% and G: 25%. The estimated Transition/Transformation bias (R) was found to be 1.51

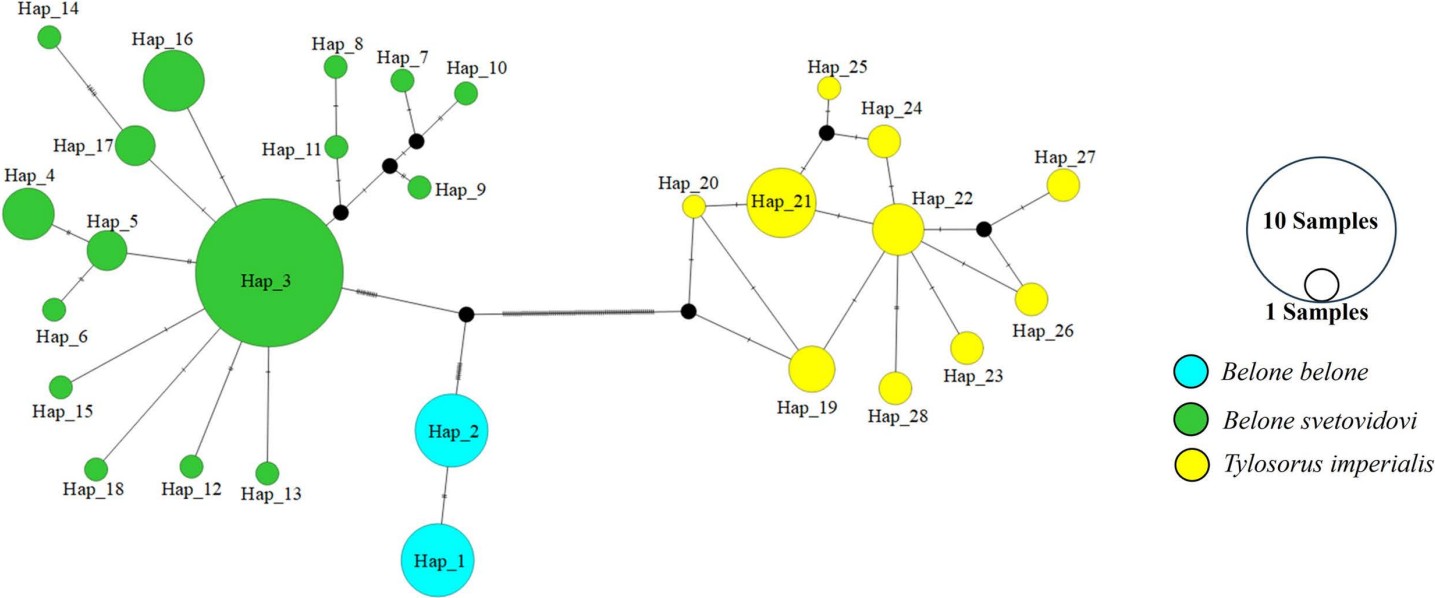

**Fig 3. Minimum spanning tree for the *COI* gene region, showing the evolutionary mutation relationships between haplotypes.**

**Table 3. The lower part of the table is given for the *COI* gene region. In bold are the within-species genetic divergence matrix and pairwise genetic distances between species that are statistically significant for the *COI* gene region. The upper part of the table is for the *12s rRNA* gene region. Intraspecific genetic divergence matrix in bold and italic letters and pairwise genetic distances between species that are statistically significant for the *12s rRNA* gene region are given in italics (\*\*\*; P < 0.001).**

| Species | *B. belone* | *B. svetovidovi* | *T. imperialis* |
|---|---|---|---|
| *B. belone* | **0.0017/0.0001** | *0.0050\*\*\** | *0.0832\*\*\** |
| *B. svetovidovi* | 0.0345\*\*\* | **0.0036/0.0016** | *0.0871\*\*\** |
| *T. imperialis* | 0.1679\*\*\* | 0.1699\*\*\* | **0.0030/0.0023** |

Twenty-three haplotypes were found from a total of 110 sequences. and there were no common haplotypes between the species (Table 4). Haplotype diversity was found to be 0.7843. The minimum spanning tree (MST) which shows the relationship between haplotypes is given in Fig 6. According to the MST analysis, a high amount of mutation was observed between the species.

The mean pairwise differentiation ($F_{ST}$) within species was found to be the lowest in the *B. belone* species and the highest in the *T. imperialis* species (Table 3). In pairwise comparison of species, the $F_{ST}$ values ranged from 0.0050 to 0.0871 (Table 3). The pairwise $F_{ST}$ values showed that some species were significantly distinct from each other (P < 0.05).

In order to gain insight into the evolutionary relationships among the species under consideration, two phylogenetic tree analyses were conducted. These were the Maximum Likelihood (ML) and the maximum parsimony (MP) analyses. The resulting trees are shown in Fig 7 and Fig 8. ML and MP trees indicated similar tree topologies. The two species of the genus *Belone*, *B. belone* and *B. svetovidovi*, were separated at the same branch, while the other species of the genus *Tylosurus*, *T. imperialis*, was clustered as a separate branch. According to ML and MP trees, *B. belone*, *B. svetovidovi* and *T. imperialis* were clustered in different sets. Also, the species *Tylosurus imperialis* differs from *Tylosurus acus* and other *Tylosurus* species at the

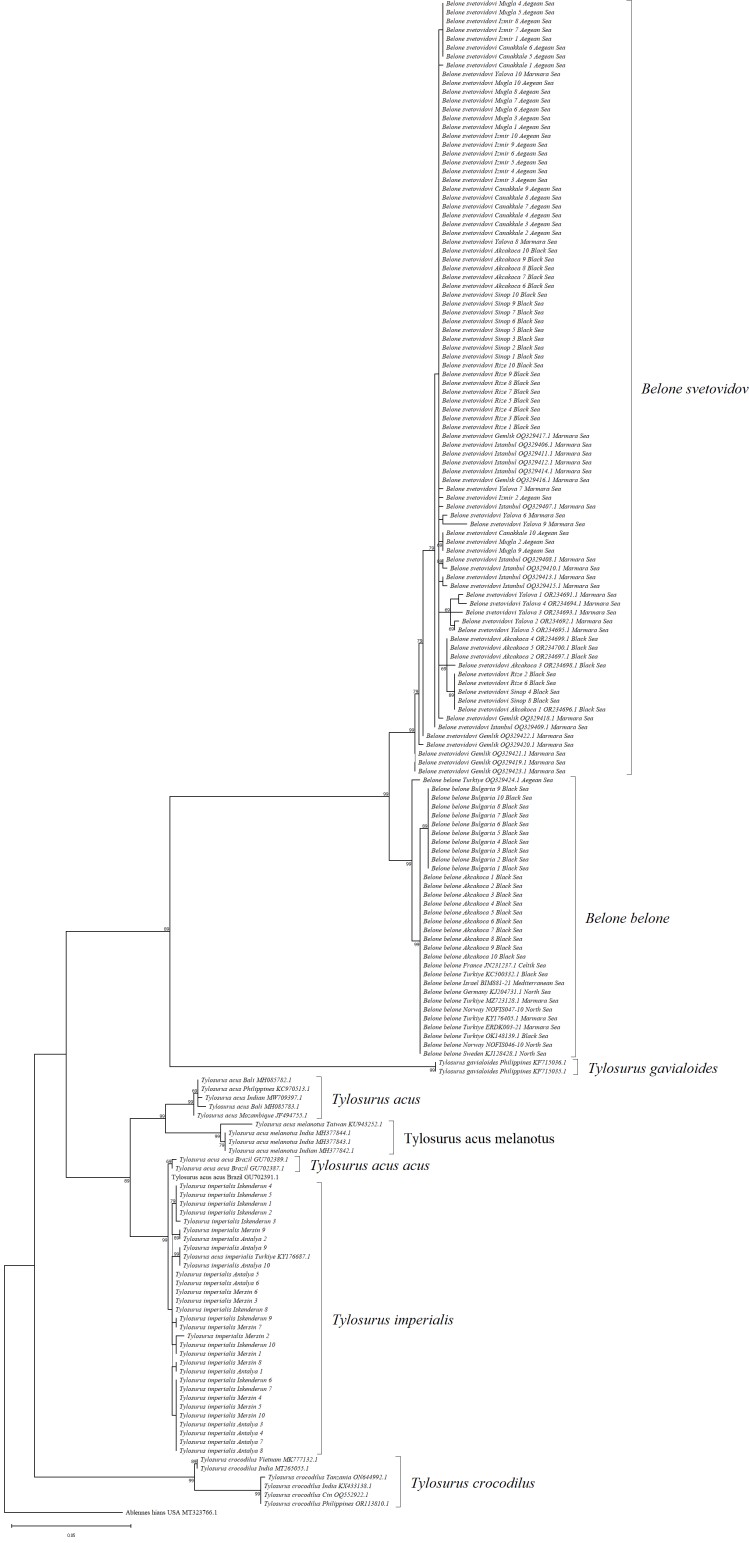

**Fig 4. Maximum Likelihood tree (ML) of needlefish species with reference samples for *COI* gene region.**

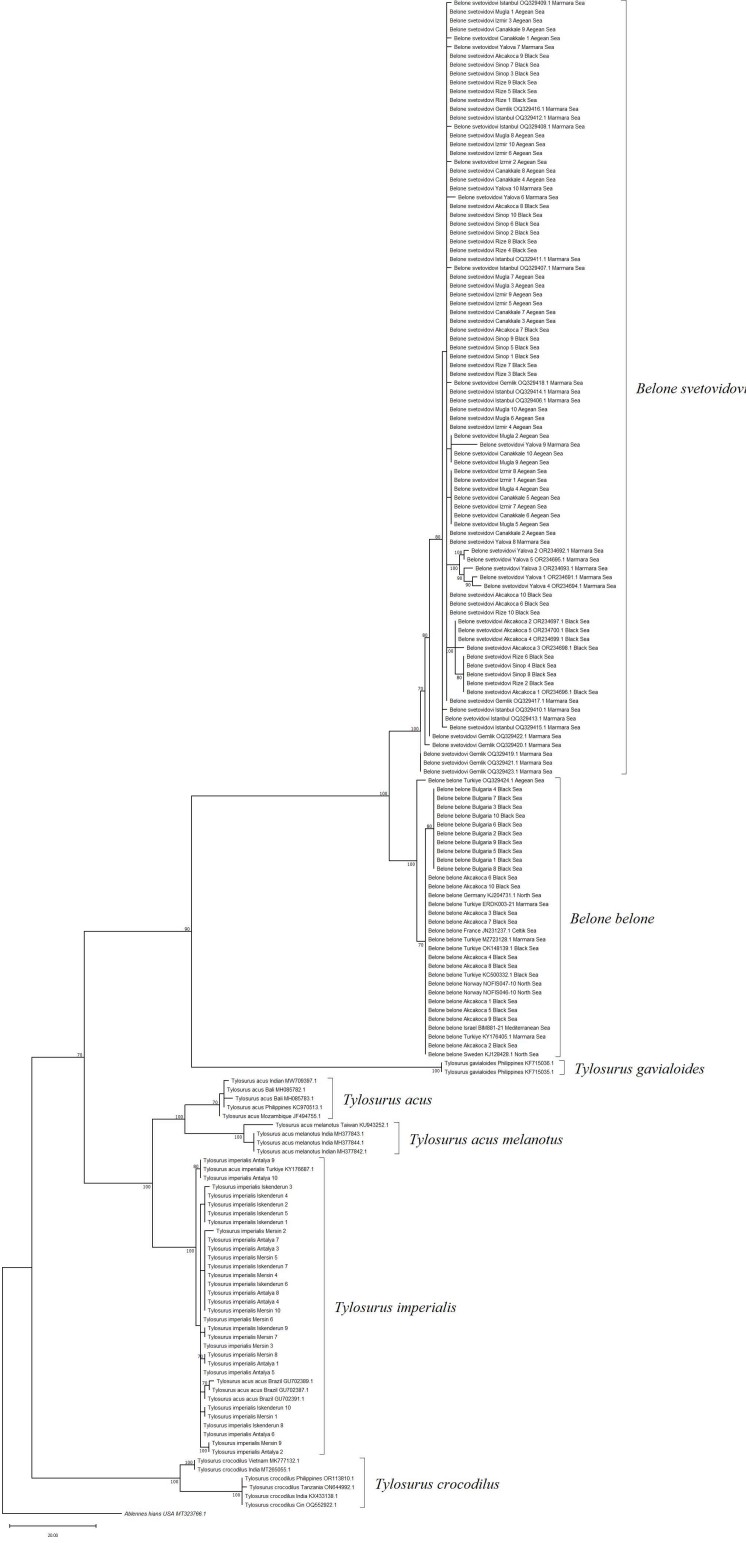

**Fig 5. Maximum Parsimony tree (MP) of needlefish species with reference samples for *COI* gene region.**

**Table 4. Haplotype number and its distribution between the species for *12s rRNA* gene region (Populatios, AKC: Akcakoca; RIZ: Rize; SIN: Sinop; YAL: Yalova, CAN: Çanakkale, IZM: Izmir, MUG: Muğla, ANT: Antalya, MER: Mersin, ISK: Iskenderun).**

| Haplotypes | *B. belone* | *B. svetovidovi* | | | | | | | *T. imperialis* | | | Total |
|---|---|---|---|---|---|---|---|---|---|---|---|---|
| | Black Sea | Black Sea | | | Aegean Sea | | | | Mediterranean Sea | | | |
| | AKC | RIZ | SIN | AKC | YAL | CAN | IZM | MUG | ISK | MER | ANT | |
| Hap_1 | 10 | – | – | – | – | – | – | – | – | – | – | 10 |
| Hap_2 | – | 9 | 9 | 10 | 9 | 1 | 2 | 2 | – | – | – | 42 |
| Hap_3 | – | 1 | 1 | – | – | – | – | – | – | – | – | 2 |
| Hap_4 | – | – | – | – | 1 | – | – | – | – | – | – | 1 |
| Hap_5 | – | – | – | – | – | 7 | 6 | 8 | – | – | – | 21 |
| Hap_6 | – | – | – | – | – | 1 | – | – | – | – | – | 1 |
| Hap_7 | – | – | – | – | – | 1 | – | – | – | – | – | 1 |
| Hap_8 | – | – | – | – | – | – | 2 | – | – | – | – | 2 |
| Hap_9 | – | – | – | – | – | – | – | – | 2 | – | – | 2 |
| Hap_10 | – | – | – | – | – | – | – | – | 1 | 9 | 9 | 19 |
| Hap_11 | – | – | – | – | – | – | – | – | 1 | – | – | 1 |
| Hap_12 | – | – | – | – | – | – | – | – | 3 | – | – | 3 |
| Hap_13 | – | – | – | – | – | – | – | – | 1 | – | – | 1 |
| Hap_14 | – | – | – | – | – | – | – | – | 1 | – | – | 1 |
| Hap_15 | – | – | – | – | – | – | – | – | 1 | – | – | 1 |
| Hap_16 | – | – | – | – | – | – | – | – | – | 1 | – | 1 |
| Hap_17 | – | – | – | – | – | – | – | – | – | – | 1 | 1 |
| Total | 10 | 70 | | | | | | | 30 | | | 120 |

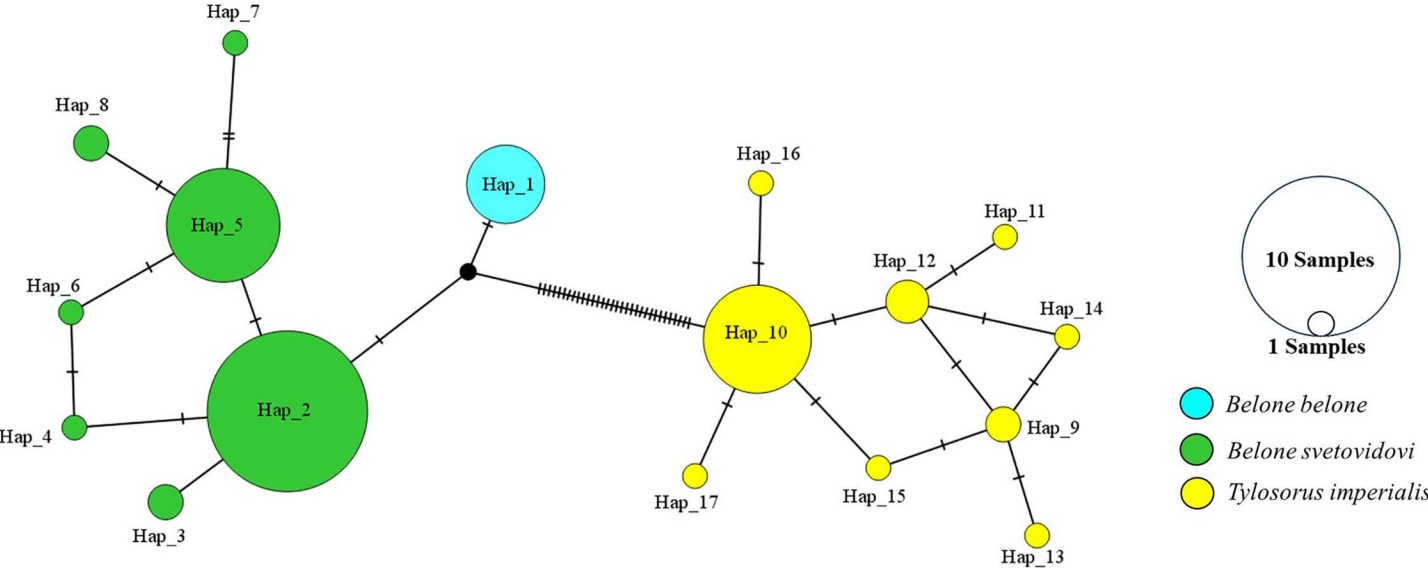

**Fig 6. Minimum spanning tree for the *12s rRNA* gene region, showing the evolutionary mutation relationships between haplotypes.**

species level by forming a separate branch with *T. imperialis* reference sequences. These results prove that the species previously recorded as *T.a. imperialis* from the Mediterranean Sea *is Tylosorus imperialis.*

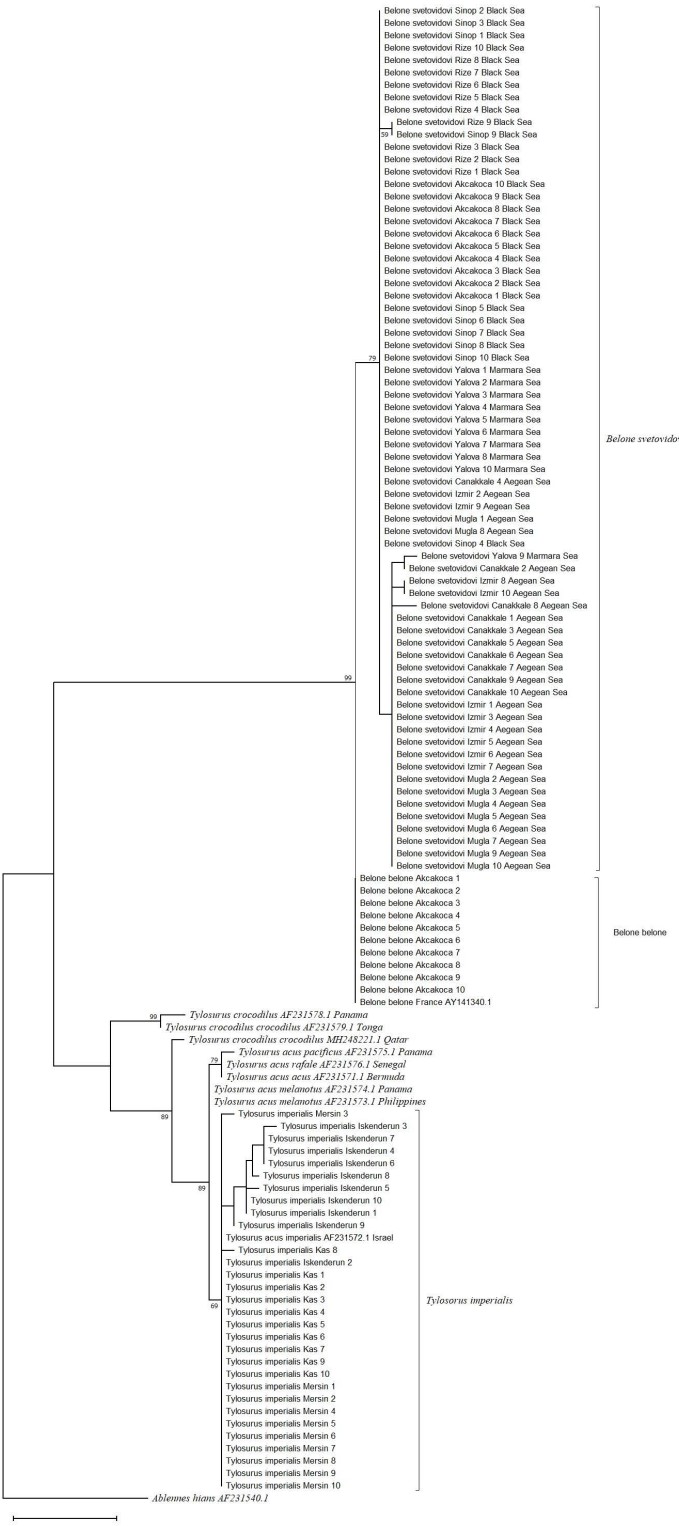

**Fig 7. Maximum Likelihood tree (ML) of needlefish species with reference samples for *12s rRNA* gene region.**

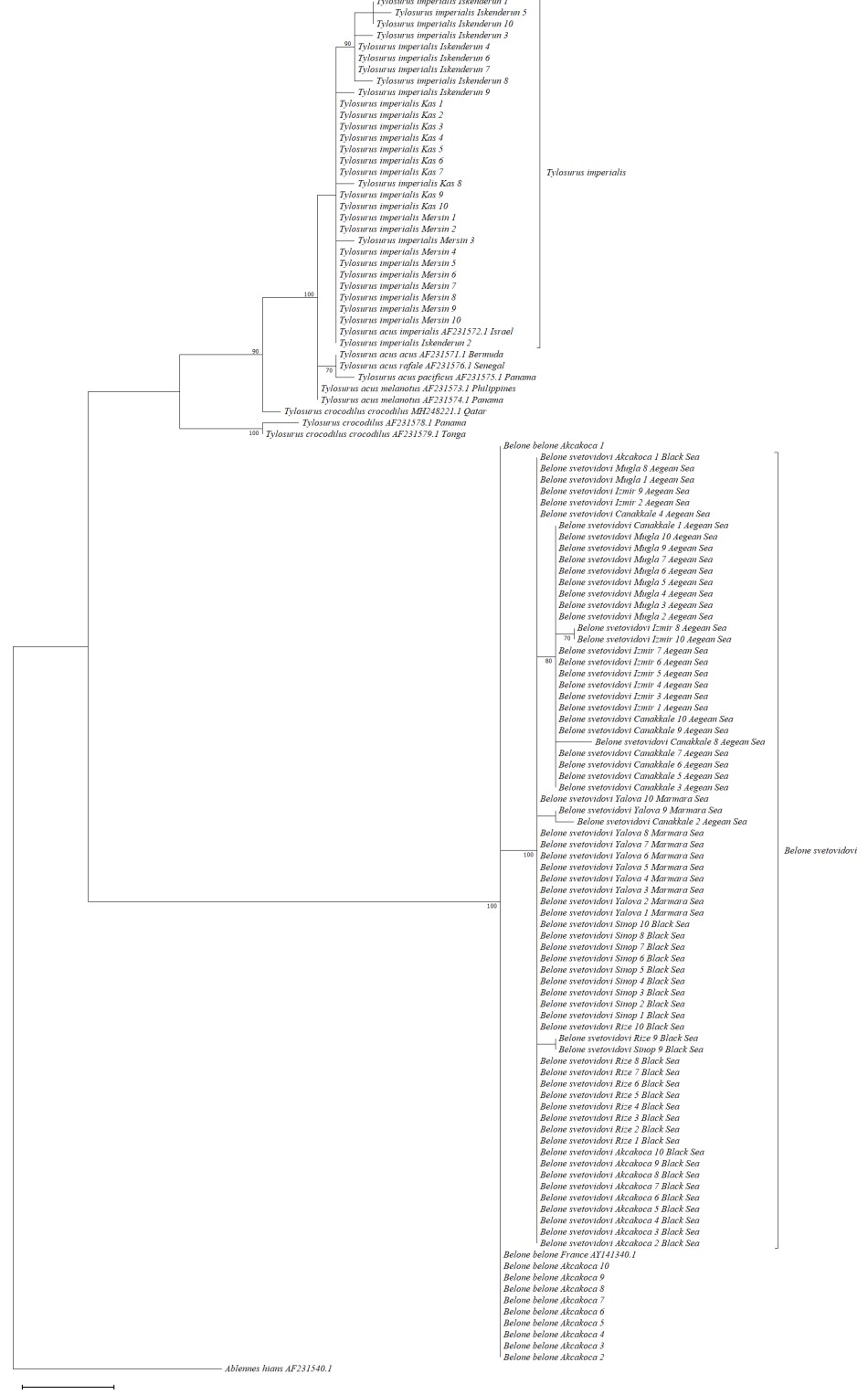

**Fig 8. Maximum Parsimony tree (MP) of needlefish species with reference samples for *12s rRNA* gene region.**

## Morphology

Discriminative morphometric and meristic characters of sampled species are given in Table 5 and Table 6. *B. belone* is morphometrically similar to *B. svetovidovi* but has a heavier, less compressed body and wider inter-orbital width. The lower jaw is a little longer than the upper jaw (Fig 9). Teeth are usually present on the vomer in specimens larger than 28 cm in Standard

**Table 5. Relative relationships of measured body proportions of needlefish collected from Turkish Marine Waters and the Bulgarian coast. N: number of fish studied, MV: mean value, SD: Standard deviation, SE: Standard error of mean value; CV: coefficient of variation.**

| Species | Relation | Sex | N | Range | MV±SE | MV±SD | CV (%) |
|---|---|---|---|---|---|---|---|
| *B. belone* | FL/TL | F + M | 50 | 0.94-0.98 | 0.957 ± 0.001 | 0.957 ± 0.007 | 0.73 |
| | SL/TL | F + M | 50 | 0.89-0.96 | 0.920 ± 0.001 | 0.920 ± 0.012 | 1.30 |
| | HL/TL | F + M | 50 | 0.12-015 | 0.134 ± 0.001 | 0.134 ± 0.008 | 5.97 |
| | O/TL | F + M | 50 | 0.02-0.03 | 0.026 ± 0.000 | 0.026 ± 0.004 | 15.38 |
| | H/HL | F + M | 50 | 0.05-0.07 | 0.059 ± 0.000 | 0.059 ± 0.002 | 3.38 |
| | h/TL | F + M | 50 | 0.01-0.04 | 0.022 ± 0.000 | 0.022 ± 0.003 | 13.63 |
| | PODL/TL | F + M | 50 | 0.81-0.87 | 0.846 ± 0.001 | 0.846 ± 0.012 | 1.42 |
| | PDL/TL | F + M | 50 | 0.65-0.71 | 0.680 ± 0.002 | 0.680 ± 0.014 | 2.05 |
| | POAL/TL | F + M | 50 | 0.81-0.88 | 0.847 ± 0.002 | 0.847 ± 0.015 | 1.77 |
| | PAL/TL | F + M | 50 | 0.63-0.70 | 0.659 ± 0.002 | 0.659 ± 0.014 | 2.12 |
| | PVL/TL | F + M | 50 | 0.49-0.55 | 0.518 ± 0.002 | 0.518 ± 0.015 | 2.89 |
| | PPL/TL | F + M | 50 | 0.11-0.18 | 0.155 ± 0.001 | 0.155 ± 0.012 | 7.74 |
| **Species** | **Relation** | **Sex** | **N** | **Range** | **MV±SE** | **MV±SD** | **CV (%)** |
| *B. svetovidovi* | FL/TL | F + M | 350 | 0.94-0.99 | 0.962 ± 0.000 | 0.962 ± 0.006 | 0.62 |
| | SL/TL | F + M | 350 | 0.90-0.97 | 0.924 ± 0.000 | 0.924 ± 0.008 | 0.86 |
| | HL/TL | F + M | 350 | 0.09-0.17 | 0.134 ± 0.000 | 0.134 ± 0.012 | 8.95 |
| | O/TL | F + M | 350 | 0.02-0.04 | 0.027 ± 0.000 | 0.027 ± 0.004 | 14.81 |
| | H/TL | F + M | 350 | 0.05-0.08 | 0.062 ± 0.000 | 0.062 ± 0.004 | 6.45 |
| | h/LH | F + M | 350 | 0.01-0.03 | 0.024 ± 0.000 | 0.024 ± 0.003 | 12.5 |
| | POD/TL | F + M | 350 | 0.81-0.91 | 0.850 ± 0.000 | 0.850 ± 0.011 | 1.29 |
| | PDL/TL | F + M | 350 | 0.65-0.76 | 0.689 ± 0.000 | 0.689 ± 0.012 | 1.74 |
| | POAL/TL | F + M | 350 | 0.80-0.91 | 0.845 ± 0.000 | 0.845 ± 0.012 | 1.42 |
| | PAL/TL | F + M | 350 | 0.63-0.74 | 0.667 ± 0.000 | 0.667 ± 0.012 | 1.80 |
| | PVL/TL | F + M | 350 | 0.41-0.57 | 0.526 ± 0.001 | 0.526 ± 0.020 | 3.80 |
| | PPL/TL | F + M | 350 | 0.12-0.19 | 0.149 ± 0.000 | 0.149 ± 0.013 | 8.72 |
| **Species** | **Relation** | **Sex** | **N** | **Range** | **MV±SE** | **MV±SD** | **CV (%)** |
| *T. imperialis* | FL/TL | F + M | 150 | 0.93-0.98 | 0.951 ± 0.000 | 0.951 ± 0.007 | 0.73 |
| | SL/TL | F + M | 150 | 0.90-0.96 | 0.926 ± 0.000 | 0.926 ± 0.008 | 0.86 |
| | HL/TL | F + M | 150 | 0.04-0.07 | 0.059 ± 0.000 | 0.059 ± 0.004 | 6.78 |
| | O/TL | F + M | 150 | 0.02-0.04 | 0.026 ± 0.000 | 0.026 ± 0.003 | 11.53 |
| | H/TL | F + M | 150 | 0.04-0.07 | 0.059 ± 0.000 | 0.059 ± 0.004 | 6.78 |
| | h/LH | F + M | 150 | 0.01-0.03 | 0.022 ± 0.000 | 0.022 ± 0.002 | 9.09 |
| | POD/TL | F + M | 150 | 0.85-0.91 | 0.877 ± 0.000 | 0.877 ± 0.010 | 1.14 |
| | PDL/TL | F + M | 150 | 0.65-0.75 | 0.680 ± 0.000 | 0.680 ± 0.010 | 1.47 |
| | POAL/TL | F + M | 150 | 0.84-0.91 | 0.872 ± 0.000 | 0.872 ± 0.009 | 1.03 |
| | PAL/TL | F + M | 150 | 0.63-0.74 | 0.672 ± 0.000 | 0.672 ± 0.011 | 1.63 |
| | PVL/TL | F + M | 150 | 0.45-0.55 | 0.489±± 0.001 | 0.489±± 0.012 | 2.45 |
| | PPL/TL | F + M | 150 | 0.12-0.17 | 0.138±± 0.000 | 0.138±± 0.010 | 7.24 |

**Table 6. Meristic characteristic of needlefish. Dorsal fin ray number (DIS), Anal ray number (AIS), Pectoral fin ray number (PIS), Number of teeth, Number of gill spines, and Number of vertebrae were counted. In addition, data on the gonad status of each specimen were taken. N: number of fish studied, MV: mean value, SD: standard deviation, SE: standard error of mean value, CV: coefficient of variation.**

| Species | Meristic characters | Sex | N | Range | MV±SE | MV±SD | CV (%) |
|---|---|---|---|---|---|---|---|
| *B. belone* | Dorsal fin ray number (DIS) | F + M | 50 | 16-19 | 17.125 ± 0.44 | 17.125 ± 1.24 | 7.24 |
| | Anal ray number (AIS) | F + M | 50 | 21-23 | 21.750 ± 0.25 | 21.750 ± 0.71 | 3.26 |
| | Pectoral fin ray number (PIS) | F + M | 50 | 12-13 | 12.750 ± 0.16 | 12.750 ± 0.46 | 3.60 |
| | Number of upper teeth (UT) | F + M | 50 | 8-10 | 8.875 ± 0.29 | 8.875 ± 0.83 | 9.35 |
| | Number of bottom teeth (UT) | F + M | 50 | 7-9 | 7.875 ± 0.29 | 7.875 ± 0.83 | 10.53 |
| | Number of gill spines (GS) | F + M | 50 | 27-39 | 32.750 ± 1.54 | 32.750 ± 4.36 | 13.31 |
| | Number of vertebrae (V) | F + M | 50 | 76-81 | 78.125 ± 0.64 | 78.125 ± 1.80 | 2.30 |
| **Species** | **Meristic characters** | **Sex** | **N** | **Range** | **MV±SE** | **MV±SD** | **CV (%)** |
| *B. svetovidovi* | Dorsal fin ray number (DIS) | F + M | 350 | 15-20 | 17.710 ± 0.13 | 17.710 ± 1.33 | 7.50 |
| | Anal ray number (AIS) | F + M | 350 | 19-23 | 22.240 ± 0.09 | 22.240 ± 0.95 | 4.27 |
| | Pectoral fin ray number (PIS) | F + M | 350 | 12-13 | 12.790 ± 0.04 | 12.790 ± 0.43 | 3.36 |
| | Number of upper teeth (UT) | F + M | 350 | 11-18 | 14.140 ± 0.18 | 14.140 ± 1.89 | 3.59 |
| | Number of bottom teeth (UT) | F + M | 350 | 12-19 | 15.620 ± 0.16 | 15.620 ± 1.61 | 2.62 |
| | Number of gill spines (GS) | F + M | 350 | 38-46 | 41.270 ± 0.20 | 41.270 ± 2.00 | 4.84 |
| | Number of vertebrae (V) | F + M | 350 | 69-86 | 77.390 ± 0.47 | 77.390 ± 4.75 | 6.13 |
| **Species** | **Meristic characters** | **Sex** | **N** | **Range** | **MV±SE** | **MV±SD** | **CV (%)** |
| *T. imperialis* | Dorsal fin ray number (DIS) | F + M | 150 | 19-26 | 23.471 ± 0.10 | 23.471 ± 1.21 | 5.15 |
| | Anal ray number (AIS) | F + M | 150 | 18-24 | 21.300 ± 0.12 | 21.300 ± 1.33 | 6.24 |
| | Pectoral fin ray number (PIS) | F + M | 150 | 12-16 | 13.203 ± 0.69 | 13.203 ± 0.76 | 5.75 |
| | Number of upper teeth (UT) | F + M | 150 | 18-48 | 30.951 ± 0.60 | 30.951 ± 6.70 | 21.64 |
| | Number of bottom teeth (UT) | F + M | 150 | 15-45 | 29.081 ± 0.63 | 29.081 ± 7.00 | 24.07 |
| | Number of gill spines (GS) | F + M | 150 | 0 | 0 | 0 | 0 |
| | Number of vertebrae (V) | F + M | 150 | 86-96 | 91.008 ± 0.19 | 91.008 ± 2.17 | 2.38 |

*B. sveovidovi* is a species very similar to *B. belone* and previously confused with *Belone belone*. Both jaws are very elongated, upper teeth are smaller and closer together (Fig 10). Although, at first glance, it differs from *B. belone* in the size and density of its beak teeth. Dorsal fin rays 16-19; Anal fin rays 20-23; Pectoral fin rays 11-13. Gill rakers 38-52 on the first arch, Vertebrae 70-86.

length (SL). Dorsal fin rays 16-20; Anal fin rays 19-23; Pectoral fin rays 11-14. Gill rakers 27-40 on the first arch, Vertebrae 75-84.

The following characteristics serve to distinguish *B. sveovidovi* from the Mediterranean garfish, *B. belone*:: The specimen was smaller and more delicate, with 13-21 teeth in a section of the middle of the upper jaw that equalled the diameter of its eye. Additionally, teeth were present on the vomer and total gill rakers on the first gill arch, which numbered 38-52. [44–46].

The bones of *T. imperialis* are not green in color as those of the *Belone* genus. Both jaws are very elongated, anterior lobes of the dorsal and anal relatively low (respectively 10.5-13.3 and 9.7-11.7 times in body length), and pectoral and pelvic relatively short (respectively 8.0-12.4 and 10.0-14.1 times in body length). There are no teeth on the vomer. There is a narrow and small black lateral keel on each side of the caudal peduncle. The caudal fin is forked, with the lower lobe much longer than the upper lobe (Fig 11). Dorsal fin rays 20-27; Anal fin rays 18-24; Pectoral fin rays 12-14. Gill rakers are absent on the arch, Vertebrae 93-96.

## Multivariate analysis

A total of 16 morphometric measurements were examined, and there were significant differences ($P < 0.001$) for all traits when analysed over all the 3 species. In the PCA, the first

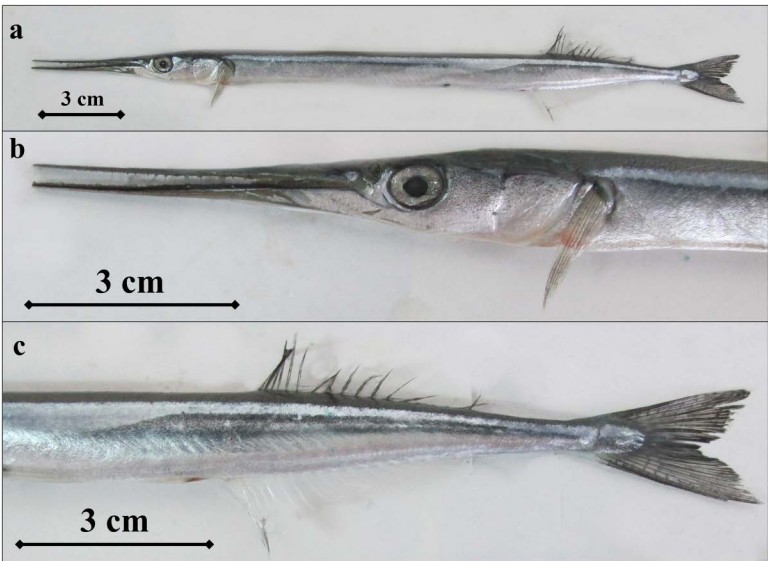

**Fig 9. B. *belone* individual (a- *B. belone's* general appearance, b- snout and head structure, c- dorsal, anus and caudal fin areas).**

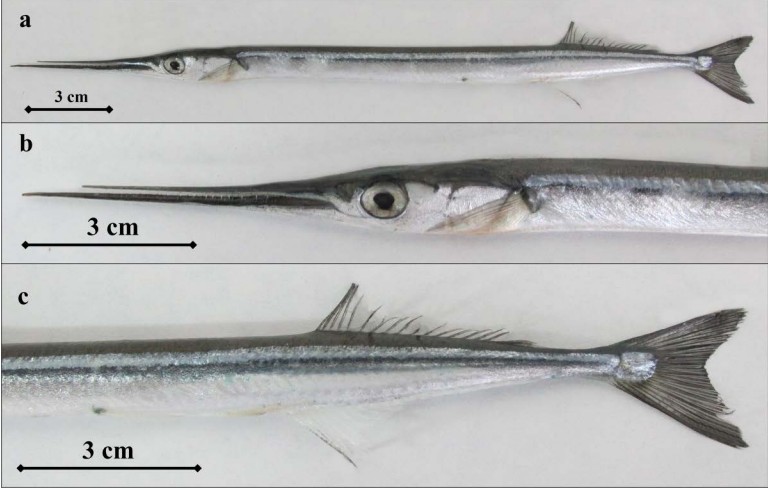

**Fig 10. B. *svetovidovi* individual (a- *B. svetovidovi* 's general appearance, b- snout and head structure, c- dorsal, anus and caudal fin areas).**

principal component, represented 38% of between group variability, was removed from the analysis. The second and third principal components produced 14% and 12% of between group variability, respectively. Examination of the contribution of each morphometric character to the second and third principal components revealed that the pectoral fin length, predorsal distance, postdorsal distance, and eye diameter, highly contributed to the morphometric differentiation of species (Fig 12).

Hierarchical canonical analysis shows that *T. imperialis* is quite distinct, with a closer morphological relationship between *B. belone* and *B. svetovidovi* (Fig 13)

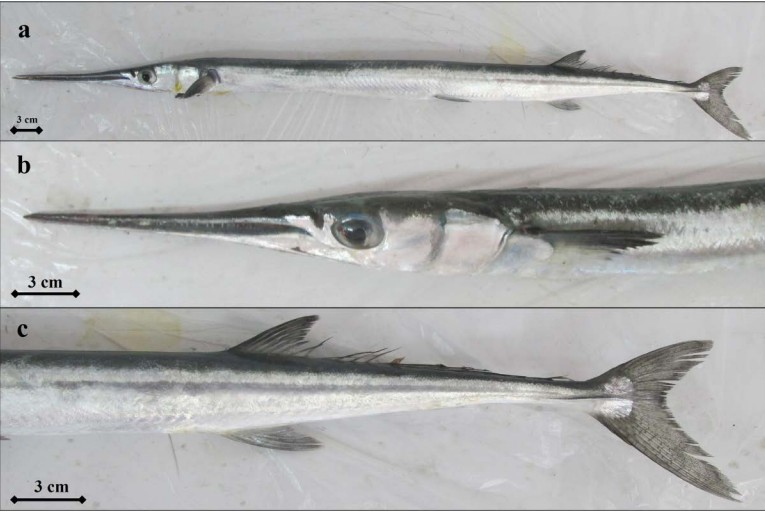

**Fig 11. *T. imperialis* individual (a- *T. imperialis's* general appearance, b- snout and head structure, c- dorsal, anus and caudal fin areas).**

**Fig 12. The contribution of each morphometric character to the second and third principal components.**

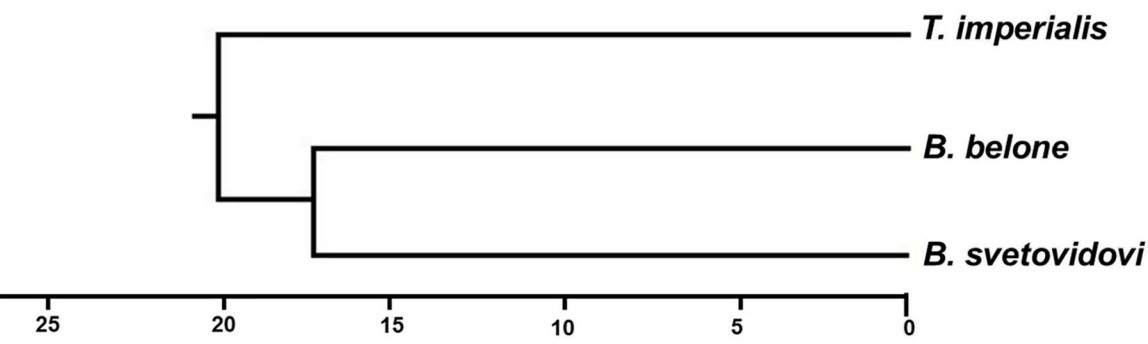

**Fig 13. Hierarchical canonical analysis of needlefish species.**

## Discussion

In the present study, systematic analysis of needlefish species was investigated by using mtDNA *COI* and *12s rRNA* gene region sequencing and morphological characters in the Turkish Marine Waters. A separate analysis of the two mitochondrial genes supported by morphological characters revealed that each species is grouped within itself.

The maximum likelihood topology revealed in our study was almost similar for two gene regions (Fig 4 and Fig 7). *T. imperialis* individuals formed a separate clade with a high bootstrap value of 99%, while *T. acus* and *T. a melanatus* subspecies clustered together to form a separate group. *B. belone* and *B. svetovidovi* species were also clustered in separate clades. The present findings for the systematic classification of three species *B. belone*, *B. svetovidovi* and *T. imperialis* found in Turkish marine waters were incongruent with the study by Banford et al. [47] studied phylogenetic relationships of the Belonid genera. Imsiridou et al. [12] studied genetic and phenotypic characters using *16s rRNA* gene region to identify *T. a. imperialis* in Thermaikos Gulf, North Aegean Sea. In the results obtained, *T. acus* and *T. a. imperialis* were grouped in the same clade. The results of our study show that *B. belone* and *B. svetovidovi* also form separate clades and are separated from each other with high bootstrap values. The maximum parsimony topology is similar to the Maximum Likelihood tree in both genes (Fig 5 and Fig 8). *T. imperialis*, *B. belone* and *B. svetovidovi* are separated from the other species by distinct clades. Moreover, ML and MP trees constructed with the *12s rRNA* gene region show that subspecies of *Tylosorus acus* such as *T. a. melenatus*, *T.a. rafale*, *T. a. acus* and *T. a. pacificus* cluster together, indicating genetic integrity.

When we look at the minimum spanning tree for both gene regions, it is seen that the species are separated from each other by many mutations (Fig 3 and Fig 6). Especially when we look at the MST of the *COI* gene region, the species are separated from each other by mutating many times. Each species has formed a separate group and is separated from each other. In the MST of the *12s rRNA* gene region, it is seen that fewer mutations occur compared to the *COI* gene region (Fig 6).

In this study, it was determined that these species differ in terms of the number of gill rakers (27-39 in *B. belone* and 38-46 in *B. svetovidovi*) and vertebrae numbers (76-81 in *B. belone* and 69-86 in *B. svetovidovi*) while there was no meristic difference in terms of the number of dorsal fin rays (16-19 in *B. belone* and 15-20 in *B. svetovidovi*), anal fin rays (21-23 in *B. belone* and 19-23 in *B. svetovidovi*), pectoral fin rays (12-13 in *B. belone* and 12-13 in *B. svetovidovi*). Dorman [10] examined the morphological characteristics of *Belone belone* and *Belone svetovidovi* species in the northern Europe between 1982 and 1985 and reported the number of gill rakers in the first arch as 30-45 for *B. belone* and 41-53 for *B. belone*,with the number of

vertebrae 76-85 for *B. belone* and 76-81 for *B. svetovidovi*. Meriç and Altun [11] indicated that the most distinct differences separating the two species (*B. belone* and *B. svetovidovi*) are the gill-rakers on the first branchial arch: *B. svetovidovi* have 39-53 gill-rakers, whereas *B. belone* has only 25-37. Moreover, the gill-rakers of *B.* svetovidovi are higher (Table 6).

Collette and Parin [8] reported that *B. belone* comprise 3 subspecies and each of which is distributed in different regions: *B. belone belone* (Linnaeus, 1761) from the north of France to the northeast of the Atlantic, *B. belone euxini* (Günther, 1866) from the Black Sea and the Sea of Azov. Fuhtermore, *B. belone gracilis* (Lowe, 1839) is found in the northwestern Atlantic and the Mediterranean waters. The taxonomic classification of these three subspecies were based on their meristic characters such as the number of vertebrae, and the number of fin rays in the dorsal fin [8]. Moreover, *Belone svetovidovi* (Collette and Parin, 1970) is found in the north-eastern Atlantic (south of Ireland, Spain, and Portugal) and the Mediterranean [8].

The most important taxonomic problem in the misidentification of *Belone belone* and *Belone svetovidovi* is that *B. belone* and *B. svetovidovi* species are phenotypically similar to each other as sibling species. On the other hand, the most important differences between *B. belone* and *B. svetovidovi* are the size and density of the beak teeth and the gills rakers number [8]. Moreover, *B. belone* has body compressed and vomerine teeth usually present in larger specimens but frequently absent in specimens less than 20 cm BL (28 cm SL). *B. belone has* comparatively large and widely spaced jaw teeth (6-15 in a distance equal to an orbit's length on the middle of the upper jaw). Moreover, *B. belone has* 75-84 vertebrae numbers, 16-20 dorsal fin rays, 19-23 anal fin rays, 11-14 pectoral fin rays,. 27-40 gill rakers on first arch. Nevertless, the body of *B. svetovidovi* is more compressed and narrower inter-orbital width. *B. svetovidovi* is characterized as 13-21 teeth within a section of the middle of the upper jaw equaling the diameter of its *eye* present on the vomer, 70-86 vertebrae numbers, 16-19 dorsal fin rays, 20-23 anal fin rays, 11-13 pectoral fin rays, 38-52 gill rakers on first gill arch [13].

Consequently, in the present study, we found that the number of gill-rakers among the meristic characters is the best for differentiation of *B. belone* from *B. svetovidovi* (Table 7).

Although these four *Tylosurus* species are morphologically similar to each other that they differ from each other in terms of the number of vertebrae. The total number of vertebrae is between 93-96 in *T. imperialis*, 75-80 in *T. crocodilus* and 77 in *T. choram* (Table 8).

**Table 7.  The meristic features of *B. belone* and *B. svetovidovi*.**

| Species | References | DFR | AFR | PFR | Teeth (Upper Jaw) | Gill Rakers | Vertebrae |
|---------|-----------|-----|-----|-----|-------------------|-------------|-----------|
| *B. belone* | Dieuzeide [48] | 16-19 | 20-22 | 12-15 | – | – | – |
| *B. belone* | Dorman [10] | 15-20 | 17-23 | 12-14 | – | 30-45 | 76-85 |
| *B. belone* | Meriç & Altun [11] | 16-19 | 20-22 | – | 5-10 | 25-37 | 76-80 |
| *B. belone* | Golani et al. [4] | 16-19 | 19-23 | 11-14 | <13 | – | – |
| *B. belone* | This study | 16-19 | 21-23 | 12-13 | – | 27-39 | 76-81 |
| *B. svetovidovi* | Dorman [10] | 16-18 | 20-23 | 11-13 | – | 42-53 | 76-81 |
| *B. svetovidovi* | Meriç & Altun [11] | 15-19 | 19-23 | – | 11-21 | 39-53 | 76-81 |
| *B. svetovidovi* | Dalyan & Eryılmaz [46] | 17 | 22 | 13 | 14 | 41 | – |
| *B. svetovidovi* | Golani et al. [4] | 16-19 | 20-23 | 11-13 | – | – | – |
| *B. svetovidovi* | This study | 15-20 | 19-23 | 12-13 | – | 38-46 | 69-86 |

*T. imperialis* has mostly been accepted as a subspecies of *T. acus imperialis* in the literature [45]. Recently, it has been considered as a provisionally accepted name until more works are conducted on *T. acus* subspecies [3].

**Table 8. The meristic features of *Tylosurus* species.**

| Species | References | DFR | AFR | PFR | Teeth (Upper Jaw) | Gill Rakers | Vertebrae |
|---|---|---|---|---|---|---|---|
| *T. acus* | Paugy [49] | 20-27 | 20-24 | – | – | 0 | 90-95 |
| *T. choram* | Collette & Parin [8] | 19-24 | 19-22 | – | – | 0 | – |
| *T. choram* | Golani et al. [4] | 20-24 | 19-22 | 12-14 | – | 0 | – |
| *T. charam* | Golani & Levi [50] | 20 | 20 | 13 | – | 0 | 77 |
| *T. crocodilus* | Collette [51] | 21-24 | 19-22 | – | – | 0 | – |
| *T. crocodilus* | Collette [52] | 20-24 | 19-22 | 12-14 | – | 0 | 75-80 |
| *T. crocodilus* | Collette [40]) | 21-23 | 18-22 | 13-15 | – | 0 | 79-84 |
| *T. a. imperialis* | Collette & Parin [8] | 23-26 | 22-23 | – | – | 0 | – |
| *T. a. imperialis* | Golani et al. [4] | 20-27 | 18-24 | 12-14 | – | 0 | – |
| *T. imperialis* | This study | 19-26 | 18-24 | 12-16 | 18-48 | 0 | 86-96 |

In conclusion, the genetic and morphological analyses showed that *Belone belone acus* and *Belone belone euxini* which are considered as the subspecies of *Belone belone* are not subspecies of the genus *Belone* and should be considered at the species level, *Belone belone*. *Belone svetovidovi* also considerably differs from *Belone belone* and should be considered as a different species. *T. acus imperialis*, which is thought to be distributed in the Mediterranean Sea, is not a subspecies of *Tylosorus acus*. *Tylosorus imperialis* differs from *Tylosorus acus* and also other *Tylosorus species* at the species level. Therefore, *T. acus imperialis* from the Mediterranean should be evaluated at the species level as *Tylosurus imperialis* due to its high genetic differentiation.

## Supporting information

**S1 Table. The sequences used in the study from NCBI and BOLD databases.**
(PDF)

**S2 Table. Genbank accession numbers of the samples used in the study.**
(XLSX)

**S3 Table. The morphological dataset used in the study.**
(XLSX)

## Author contributions

**Conceptualization:** Cemal Turan.

**Data curation:** Cemal Turan, Petya Pavlova Ivanova, Servet Ahmet Doğdu, Deniz Ergüden, Violin Stoyanov Raykov, Maria Yankova, Deniz Yağlioğlu, Ayşegül Ergenler.

**Formal analysis:** Cemal Turan, Servet Ahmet Doğdu, Deniz Ergüden, Deniz Yağlioğlu.

**Investigation:** Cemal Turan, Servet Ahmet Doğdu.

**Methodology:** Cemal Turan, Petya Pavlova Ivanova, Servet Ahmet Doğdu.

**Project administration:** Cemal Turan.

**Resources:** Cemal Turan, Servet Ahmet Doğdu, Deniz Yağlioğlu.

**Supervision:** Cemal Turan.

**Visualization:** Cemal Turan, Petya Pavlova Ivanova.

**Writing – original draft:** Cemal Turan, Petya Pavlova Ivanova, Servet Ahmet Doğdu, Deniz Ergüden, Violin Stoyanov Raykov, Maria Yankova, Deniz Yağlioğlu, Ayşegül Ergenler.

**Writing – review & editing:** Cemal Turan, Petya Pavlova Ivanova, Servet Ahmet Doğdu, Deniz Ergüden, Violin Stoyanov Raykov, Maria Yankova, Deniz Yağlioğlu, Ayşegül Ergenler.

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
