## [Decision Letter · Decision Letter 0]

25 Sep 2024

PONE-D-24-33951Systematic Identification of Needle Fish (Belonidae) Species using Molecular Genetic and Morphological Markers in the Mediterranean SeaPLOS ONE

Dear Dr. Turan,

Thank you for submitting your manuscript to PLOS ONE. After careful consideration, we feel that it has merit but does not fully meet PLOS ONE’s publication criteria as it currently stands. Therefore, we invite you to submit a revised version of the manuscript that addresses the points raised during the review process. Please carefully read and follow all the reviewers comments and suggestions. Some of them are essential for the persecution of this editorial process, but I believe this manuscript could be improved by your work in rewriting and elucidates several parts. I agree with the comment about the use of just a mitochondrial marker in your molecular analyses, affecting your results, this should be considered in the discussion and conclusion section as a limitation of this study, which certainly needs implementations from this point of view.

We look forward to receiving your revised manuscript. Have a nice work!

Kind regards,

Marco Albano, Ph.D.

Academic Editor

PLOS ONE

- Morphological and genetic characteristics of garfish Belone belone (L., 1760) (Belonidae, Teleostei) population from the southern Bulgarian Black Sea coast (DOI:10.3897/natureconservation.54.113071)

-Family Belonidae Bonaparte 1832 (https://www.calacademy.org/sites/default/files/assets/docs/belonidae.pdf)

-Existence of Belone svetovidovi Collette & Parin, 1970 in the Marmara Sea and Black Sea Coasts of Türkiye (doi: 10.28978/nesciences.1331296)

(among others)

In your revision ensure you cite all your sources (including your own works), and quote or rephrase any duplicated text outside the methods section. Further consideration is dependent on these concerns being addressed.

4. We suggest you thoroughly copyedit your manuscript for language usage, spelling, and grammar. If you do not know anyone who can help you do this, you may wish to consider employing a professional scientific editing service.

“This research was supported by the Scientific & Technological Research Council of Turkey (TUBITAK-121N777) and Bulgarian Academy of Sciences (BAS-IC-TR/1/2022-2023). The authors thank to TUBITAK for their supports.”

“This research was supported by the Scientific & Technological Research Council of Turkey (TUBITAK-121N777) and Bulgarian Academy of Sciences (BAS-IC-TR/1/2022-2023). The authors thank to TUBITAK for their supports.”

“This research was supported by the Scientific & Technological Research Council of Turkey (TUBITAK-121N777) and Bulgarian Academy of Sciences (BAS-IC-TR/1/2022-2023). The authors thank to TUBITAK for their supports.”

7. In the online submission form you indicate that your data is not available for proprietary reasons and have provided a contact point for accessing this data. Please note that your current contact point is a co-author on this manuscript. According to our Data Policy, the contact point must not be an author on the manuscript and must be an institutional contact, ideally not an individual. Please revise your data statement to a non-author institutional point of contact, such as a data access or ethics committee, and send this to us via return email. Please also include contact information for the third party organization, and please include the full citation of where the data can be found.

8. We note that Figures 2, 9, 10 and 11 in your submission contain copyrighted images. All PLOS content is published under the Creative Commons Attribution License (CC BY 4.0), which means that the manuscript, images, and Supporting Information files will be freely available online, and any third party is permitted to access, download, copy, distribute, and use these materials in any way, even commercially, with proper attribution. For more information, see our copyright guidelines: http://journals.plos.org/plosone/s/licenses-and-copyright.

1. You may seek permission from the original copyright holder of Figures 2, 9, 10 and 11 to publish the content specifically under the CC BY 4.0 license.

9. We note that Figure 1 in your submission contain [map/satellite] images which may be copyrighted. All PLOS content is published under the Creative Commons Attribution License (CC BY 4.0), which means that the manuscript, images, and Supporting Information files will be freely available online, and any third party is permitted to access, download, copy, distribute, and use these materials in any way, even commercially, with proper attribution. For these reasons, we cannot publish previously copyrighted maps or satellite images created using proprietary data, such as Google software (Google Maps, Street View, and Earth). For more information, see our copyright guidelines: http://journals.plos.org/plosone/s/licenses-and-copyright.

10. Please include captions for your Supporting Information files at the end of your manuscript, and update any in-text citations to match accordingly. Please see our Supporting Information guidelines for more information: http://journals.plos.org/plosone/s/supporting-information.

Reviewers' comments:

Reviewer's Responses to Questions

**Comments to the Author**

1. Is the manuscript technically sound, and do the data support the conclusions?

Reviewer #1: Partly

Reviewer #2: Yes

Reviewer #3: Partly

2. Has the statistical analysis been performed appropriately and rigorously? 

Reviewer #1: N/A

Reviewer #2: Yes

Reviewer #3: Yes

3. Have the authors made all data underlying the findings in their manuscript fully available?

Reviewer #1: Yes

Reviewer #2: Yes

Reviewer #3: No

4. Is the manuscript presented in an intelligible fashion and written in standard English?

Reviewer #1: No

Reviewer #2: No

Reviewer #3: No

5. Review Comments to the Author

Reviewer #1: Manuscript need to be checked by a native English Speaker. for details, please also see the attached file.

species name must be italicized.

what about topotyps and paratypes in discovered/renamed species?

Reviewer #2: Dear Authors,

the work presented in this study is particularly interesting because it offers valuable tools for the identification of species within the Belonidae family, which are known to be very similar to each other. The genetic and morphological approaches used here are crucial for clarifying the taxonomic status of these species, particularly in regions like the Mediterranean and Black Seas where they are distributed. However, several aspects of the manuscript need to be addressed to improve its clarity and overall quality (please refer to the comments in the attached PDF for detailed feedback). Some sections of the paper appear to be unclear and repetitive, which could confuse readers and detract from the strength of your findings. Additionally, the English language used throughout the manuscript requires revision to enhance readability and ensure that the scientific content is communicated effectively. You also need to consider and discuss Ablennes hians. In summary, while the study contributes valuable insights into the taxonomy of Belonidae species, careful attention to the points mentioned above will greatly improve the manuscript’s impact.

Reviewer #3: Comments to the Redactor and Authors

I thank the journal authorities and the Editor for handing me an opportunity to review a manuscript: “Systematic Identification of Needle Fish (Belonidae) Species using Molecular Genetic and Morphological Markers in the Mediterranean Sea”.

Despite a number of serious concerns, I am confident that this research is competent. Nevertheless, I also suggest that the scientific significance of this manuscript is not appropriate for the journal PLOS ONE. In this manuscript authors try to clarify the taxonomic relationships between species within the genera Belone and Tylosaurus, using markers from the mitochondrial genome and morphological traits.

While the authors have used a significant number of specimens, the results they present are not of sufficient interest for a broad-ranging, multidisciplinary journal like PLOS ONE. Instead, I recommend that this work be transferred to a more specialized, thematic ichthyological journal that would be better suited to this manuscript.

General comments:

Authors need to significantly improve the quality of the manuscript, because nowadays it is not suitable for reading and understanding.

1. First of all, the manuscript is full of misleading typos, missing punctuation marks, and grammar mistakes.

2. Second, the English in this manuscript should be significantly improved.

3. Third, the quality of figures (maybe due to transfer to PDF) is low; it is impossible to read the legends on the figures.

4. Fourth, I did not find the NCBI numbers for COI and 12S rRNA sequences in the manuscript.

5. Fifth, today, and this is the rule, in taxonomic studies, it is necessary to use data not only from the mitochondrial genome but also from the nuclear genome. This is because the mitochondrial genome has predominant maternal inheritance.

Minor comments:

1. The manuscript title implies study of specimens from the Mediterranean Sea (not the Black Sea).

2. Along the title and text: Please change "needle fish" to "needlefish".

3. Gene names should be italicized in the manuscript.

4. Sub-species should be changed to subspecies in the text.

5. If a name of genera is mentioned once, then it should be abbreviated further in the text.

6. It is necessary to add scales to Figures 9–11. The quality of Figures 4, 5, and 7–8 is poor, making it impossible to see anything in the phylogenetic trees. Figure 13 is unclear as to what it can show in evolutionary terms?

6. PLOS authors have the option to publish the peer review history of their article (what does this mean? ). If published, this will include your full peer review and any attached files.

**Do you want your identity to be public for this peer review?** For information about this choice, including consent withdrawal, please see our Privacy Policy .

Reviewer #1: No

Reviewer #2: **Yes: ** Francesco Tiralongo

Reviewer #3: No

---

## [Author Response · Author response to Decision Letter 1]

13 Nov 2024

Referee and editor comments on our article have been carefully reviewed. The corrections made according to the comments are given in the “Response to Reviewers” file.

---

## [Decision Letter · Decision Letter 1]

26 Nov 2024

Systematic Identification of Needlefish (Belonidae) Species using Molecular Genetic and Morphological Markers in the Mediterranean and Black Seas

PONE-D-24-33951R1

Dear Dr. Turan,

We’re pleased to inform you that your manuscript has been judged scientifically suitable for publication and will be formally accepted for publication once it meets all outstanding technical requirements.

Kind regards,

Marco Albano, Ph.D.

Academic Editor

PLOS ONE

Additional Editor Comments (optional):

Dear Authors,

thanks to have seriously interact with the reviewers during this editorial process, solving almost all their requests, or reasonably arguing different point of views which comes out.

Best regards

Reviewers' comments:

Reviewer's Responses to Questions

**Comments to the Author**

1. If the authors have adequately addressed your comments raised in a previous round of review and you feel that this manuscript is now acceptable for publication, you may indicate that here to bypass the “Comments to the Author” section, enter your conflict of interest statement in the “Confidential to Editor” section, and submit your "Accept" recommendation.

Reviewer #2: All comments have been addressed

2. Is the manuscript technically sound, and do the data support the conclusions?

Reviewer #2: Yes

3. Has the statistical analysis been performed appropriately and rigorously? 

Reviewer #2: Yes

4. Have the authors made all data underlying the findings in their manuscript fully available?

Reviewer #2: Yes

5. Is the manuscript presented in an intelligible fashion and written in standard English?

Reviewer #2: Yes

6. Review Comments to the Author

Reviewer #2: Dear authors, I think that now, in its current form, the article can be accepted. All comments and suggestions were followed.

7. PLOS authors have the option to publish the peer review history of their article (what does this mean? ). If published, this will include your full peer review and any attached files.

**Do you want your identity to be public for this peer review?** For information about this choice, including consent withdrawal, please see our Privacy Policy .

Reviewer #2: No

---

## [Editor Report · Acceptance letter]

PONE-D-24-33951R1

PLOS ONE

Dear Dr. Turan,

I'm pleased to inform you that your manuscript has been deemed suitable for publication in PLOS ONE. Congratulations! Your manuscript is now being handed over to our production team.

Kind regards,

on behalf of

Dr. Marco Albano

Academic Editor

PLOS ONE